# Optimisation of 3D Printing for Microcellular Polymers

**DOI:** 10.3390/polym15193910

**Published:** 2023-09-27

**Authors:** Christian Andrew Griffiths, Andrew Rees, Adam Morgan, Feras Korkees

**Affiliations:** Faculty of Science and Engineering, Swansea University, Swansea SA1 8EN, UK; andrew.rees@swansea.ac.uk (A.R.); adam.j.morgan@swansea.ac.uk (A.M.); f.a.korkees@swansea.ac.uk (F.K.)

**Keywords:** additive manufacturing, 3D printing, fused filament fabrication, fused deposition modelling, lightweight PLA

## Abstract

Polymers are extensively used in various industries due to their versatility, durability and cost-effectiveness. To ensure functionality and longevity, polymer parts must have sufficient strength to endure external forces without deformation or breakage. Traditional approaches to increasing part strength involve adding more material; however, balancing strength to weight relationships is challenging. This paper explorers the viability of manufacturing lightweight components using a microcellular foaming polymer. Microcellular foaming has emerged as a helpful tool to achieve an optimal strength-to-weight ratio; offering advantages such as lightweight, improved mechanical properties, reduced material usage, better insulation and improved cost-effectiveness. It can also contribute to improved fuel efficiency and reduced carbon emissions, making them environmentally favourable. The combination of additive manufacturing (AM) and microcellular foaming has opened new possibilities for design innovation. This text highlights the challenges and efforts in incorporating foaming techniques into 3D printing processes, specifically fused filament fabrication (FFF). This study reveals that microcellular polymers are a viable option when balancing part strength and weight. The experiments completed during the formulation of this paper demonstrated that lightweight LW-PLA parts were significantly lighter than standard PLA parts and that a design of experiments approach can be used to optimise strength properties and provide insights into optimising manufacturability. Microcellular polymers present an opportunity for lighter and stronger 3D printed parts, offering potential energy and material savings for sustainable manufacturing practices.

## 1. Introduction

Polymers offer versatility, durability and affordability, making them essential across industries, from packaging to automotive. Robust plastic components are vital for function and longevity, necessitating resistance to external forces without deformation. Additionally, lightweight polymers enhance fuel efficiency, aiding the environment. Processing techniques are optimised to improve strength, durability, and sustainability. Traditionally, augmenting part strength meant more material, yet harmonising strength and lightness is complex for lightweight elements. Optimal strength-to-weight balance emerges through design and simulation. Polymer choice markedly affects both attributes, influenced by processing and application needs. Flexibility, stiffness, toughness, thermal stability and impact resistance vary based on processing. Ultimately, product design considers application, method and cost.

Foam polymer is an evolving lightweight structural material that exhibits remarkable characteristics such as high specific strength, exceptional energy absorption, damping and thermal properties, all coupled with a low density (approximately 10–15 times less dense than its original volume) [1,2,3,4,5,6,7]. In the 1980s, the Massachusetts Institute of Technology (MIT) introduced microcellular processing to the polymer industry. The objective was to reduce material usage, decrease the weight of final parts and modify material properties by incorporating small spherical cells into polymer-based products [8]. The initial publications and theses resulting from this research laid the foundation for the technology, proving its concept and advancing fundamental theories. The original work focused on batch processing and extrusion, leading to the granting of the first U.S. patent in 1984 [9]. Commercialisation of microcellular technology began in 1998 when Axiomatics Corp developed the first reciprocating screw microcellular injection moulding (MIM) machine. Before microcellular technology, conventional foam was utilized to create polymer parts with a cellular structure. However, the limitations of cell densities and poor mechanical properties were overcome through MIM. Microcellular foaming processes have the capability to produce smaller cell structures compared to regular chemical blowing agent methods [10]. While chemical blowing agents typically achieve cell structures as small as 250 mm, microcellular foaming can produce cells in the range of 3–100 mm [11]. MIM has also demonstrated its ability to reduce polymer content, decrease the weight of final parts and lower processing energy requirements. Various processing variables, such as injection speed, gate flow resistance, blowing agent content, melt flow rate, and talc addition, have been investigated in the context of MIM [12,13,14,15]. The unique structure of microcellular foamed polymers offers several advantages over traditional solid polymers, including the following:

Lightweight: Microcellular polymers are much lighter than traditional solid polymers, making them ideal for applications where weight reduction is important.

Improved Mechanical Properties: Microcellular polymers have improved mechanical properties, including increased stiffness, toughness and impact resistance. These properties make them ideal for many products.

Reduced Material Usage: Microcellular polymers use less material than traditional polymers to achieve the same level of strength and durability. This makes them an environmentally friendly choice as less material is used.

Better Insulation: The air bubbles in microcellular polymers act as insulators, making them ideal for use in applications where insulation is important.

Cost-effective: Microcellular foaming is a cost-effective manufacturing technique as it allows for the use of less material, resulting in lower costs for manufacturers.

In recent times, additive manufacturing (AM) of polymers has unlocked a vast array of possibilities for design innovation, enabling the creation of intricate and complex shapes [16]. A growing number of industries are embracing AM technologies to expedite product development while enhancing cost-effectiveness [17]. Through strategic material modification, process optimisation and print optimisation, polymers, blends and composites [18] have evolved to exhibit novel properties that can meet or surpass the performance of traditional products [19]. This has paved the way for exciting opportunities in 3D printed products.

The fusion of AM and microcellular foaming is a compelling route for designers. AM’s appeal lies in crafting intricate structures with ease [20]. Utilizing DLP, polymer foams with controlled porosity are additively manufactured using expandable microspheres. This technique creates resilient bio-inspired materials, sustaining modulus and energy dissipation, even during cyclic loading. In addition, these 3D-printed foams demonstrate sustained values of modulus and energy dissipation, even under repeated loading at large deformations [21]. Applications of this thermoplastic are already being utilised in fields such as aviation [22,23] and robotics [24,25] for their light weight and ability to construct complex structures.

Another approach involves the production of cellular thermoplastic structures with multiscale porosity, achieved through an integrated approach of 3D printing using fused deposition modelling (FDM). This method enables the creation of 3D porous structures that can be tailored for various applications, including drug delivery, energy storage, microfluidics and tissue engineering, where tuneable multiscale porosity is highly sought after [26]. The choice of unit cell size is crucial in these processes, as smaller cells offer advantages in applications requiring high strength, energy absorption and dissipation, while larger unit cells can be utilized for superior damping performance. These materials find potential use in applications such as protective equipment, shoe midsoles and vibration dampers [27].

Despite the progress made and the promising advantages arising from the combination of 3D printing and foaming, this technique has yet to be applied to any specific application. Nonetheless, the potential for innovation is an area of research and development [28].

Fused filament fabrication (FFF), a popular 3D printing technique, is widely used for manufacturing plastic parts. It offers cost-effectiveness and efficiency in the production process. FFF provides extensive design flexibility, allowing designers to create intricate shapes and structures. However, the production of foamed structures using FFF remains challenging due to the time-consuming process and the difficulty of creating internal micro-porous structures using existing printing techniques [29].

With the growing popularity of FFF in both industrial applications and hobbyist settings, there is increased awareness regarding the materials used and their sustainability [30]. Research has shown that 3D printing with foam filaments can achieve comparable results to traditional methods, using up to 60% less polymer volume. Furthermore, the possibility of recycling makes the process more environmentally friendly than using natural polymer filaments [31].

Efforts to address environmental concerns in 3D printing include designing experiments that focus on materials, energy consumption, production and waste generation [32]. It has been demonstrated that reductions in energy consumption and waste contribute to sustainable long-term product development in this technology [33]. Recent research has shown that printer and material modifications can significantly reduce polymer consumption from 55.8% to 56.4% and decrease power consumption from 29% to 38% [34].

To achieve the goal of making plastic parts both strong and lightweight, a comprehensive approach is required. This involves utilising advanced materials, optimising design and simulation tools and employing efficient processing techniques. Ongoing research on microcellular foamed polymers for 3D printing is expected to further expand the range of applications for this technology by advancing materials and manufacturing processes.

The objective of this paper is to reduce part weight using a design of experiments approach when printing with a lightweight (LW-PLA) filament, incorporating a foaming agent. In addition to printing LW test parts, the study focused on evaluating material properties such as tensile strength (σ), stiffness (k) and strain to failure (εf). The paper is structured as follows: the experimental section is presented first, followed by the results and subsequent discussion. Finally, conclusions are drawn regarding the influence of the manufacturing process on weight savings and the observed relationship with material strength properties.

## 2. Experimental Design

### 2.1. 3D Printer

The printer used for testing was a modified Ender-3 Pro ‘Creality Ender’ series (Shenzhen Creality 3D Technology Co., Ltd., Shenzhen, China). This printer was modified to rectify a range of flaws from the original design which included using a Creality Sprint direct drive extruder for more accurate extrusion of material, a secondary lead screw and stepper motor supporting the Z axis to remove sag in the gantry, stronger bed levelling springs for sturdier print bed support, a polyetherimide (PEI)-coated print bed for greater part adhesion, the addition of a CR-Touch sensor for bed levelling and tramming capabilities and finally, a touch screen control panel for improved bed-levelling visualisation. To ensure consistency, the slicer used throughout was Ultimaker CURA version 4.12 with default settings for each printer, all tests used 0.4 mm diameter brass nozzles and the bed was re-levelled between each batch of prints. The control filament used was black economy polylactic acid (PLA) supplied by colorFabb (Limburg, The Netherlands). The filament used to investigate the potential polymer savings provided by foaming agents within the filament was colorFabb Natural LW-PLA. The ambient conditions were maintained by a thermostatically controlled electrical heater set to 25 °C.

### 2.2. Test Part and Printer Settings

The test part used is ISO 527-2-1A [35]. ISO 527-2 is an international standard for determining the tensile properties of reinforced and non-reinforced plastics. The model is automatically imported flat to the base on the centre of the build plate. G-code for the test part was generated using the settings in Table 1. Ten parts were produced using this setting using colorFabb black economy PLA. An additional 45 parts were produced using the colorFabb Natural LW-PLA; this is detailed in Section 2.4.

### 2.3. Tensile Testing

A Hounsfield electro-mechanical test machine was used with a load-cell of 25 kN. Tests were performed at two different speeds (strain rates) with 1 mm/min in the first stage up to 0.3% elongation followed by speed of 5 mm/min until failure. An Epsilon E97197 extensometer (3542-025 M-010-ST, Jackson, WY, USA) was used to measure the strain and it was directly attached to the sample during testing. Elastic modulus (stiffness) values were estimated from the slope of the stress/strain curve as close as possible to the strain interval between ε_1_ = 0.05% and ε_2_ = 0.25%.

The testing was focused of the following response variables:

Tensile strength (σ) defined as force per unit area, which is associated with the amount of tensile stress a material can withstand before breaking.

Strain to failure (ε_f_) gives the measure of how much the specimen is elongated to failure.

Stiffness (k) is the extent to which an object resists deformation in response to an applied force.

### 2.4. Design of Experiments

To investigate the behaviour of microcellular foamed polymers manufactured using the FFF process on part, this experimental research was focused on response variables of part weight (W), build time (t_B_) and the tensile properties of σ, ε_f_, k and σ-ε. Part production is highly dependent on control factors and therefore affected by layer height (L_h_), temperature (T) and speed (S). As three factors at three levels were considered, a Taguchi L9 orthogonal array (OA) was selected as shown in Table 2. The control factors were originally based on the capabilities of the printer and the minimum and maximum values within the recommended range for the material. However, pre-trials identified the range of where quality prints were possible. The L_h_ settings used in this research were from 0.18 to 0.28 mm. This range of heights was considered to replicate common layer heights that are typically used when prototyping parts using FFF printers. The F was reduced to 65% to accommodate the expansion of material as the LW-PLA foams. The recommended T range for the material was 195–250 °C, but a T of 240–260 °C was identified as optimum for the build. The speed range of 40–50 mm/s was selected based on already established optimal flow rates for PLA-like filaments. For each combination of controlled parameters for the selected L9 OAs (as presented in Table 2), 5 runs were performed and a total of 45 experimental trials were carried out.

## 3. Results

### 3.1. Microcellular Foaming

To establish whether the foaming process was achieved with all nine runs, the test parts were cut in two and inspected using a Keyence VHX1000 high depth-of-field light microscope. To avoid plastic deformation of the microcellular structure, the samples were scored along the cross-section to localise the fracture line. The samples were then submerged in liquid nitrogen for two minutes before the fracture was performed. For all runs, it can be observed that the microcellular structure has been achieved through the extrusion process, as shown in Figure 1. Due to the volume of bubbles, it is not possible to discern the difference between the sum of bubbles and the population densities. However, for all runs, the diameter of the bubble structure ranges between 40 and 80 μm. In the following section, results from the mechanical testing will identify the microcellular performance for part weight and mechanical strength properties.

### 3.2. Part Weight

#### 3.2.1. Mean Results

In this study, an L16 OA was employed. For each combination of controlled parameter carried out, weight measurements were obtained. The weights from the nine experiments are presented in Table 3 and Figure 2. The results show that the three factors influence the weight of the parts. In particular, experiment 5 has the lowest mean W of 8.152 g and experiment 2 has the highest at 8.5 g. This is a difference of 4.0%. Out of all the experiments, the distribution of experiments 2 and 8 have the lowest variation (SE Mean 0.009). The pooled average of all nine experiments can be used to test the hypothesis (H_1_) that the LW-PLA parts are lighter than standard PLA parts. The paired *t*-test result shows that the *p*-value is below 0.05 (Table 4), confirming that there is a significant difference between the means and H_1_ is correct.

#### 3.2.2. Print Parameters’ Contribution to Optimum Weight

Based on the experimental results, an analysis of variance (ANOVA) was performed in order to assess the contribution of each processing parameter to the resulting weight. Table 5 shows the rank importance and percentage contribution of each parameter and also includes ranks based on δ statistics, which compare the relative magnitude of effects, where the δ statistic is the highest minus the lowest average for each factor. The results for weight show that L_h_ has the highest contribution of 2.84%. However, each *p* value for each of the factors above 0.05 confirm that they are not significant (Table 5). The factor levels that result in a low weight are L_h_ 0.28 mm, T 260 °C and S 50 mm/s. Using the predicted result for this setting, it is possible to produce a part with a weight of 8.15 g. This is 21.22% below the average weight of a test part made from original PLA.

In Section 3.2.2, it was shown that experiment 5 has the lowest mean weight and experiment 2 has the highest, with a difference of 4.0% observed. The high weight observed in experiment 2 uses settings 1, 2 and 2 for L_h_, T and S, respectively (Table 2). The results in Table 5 confirm that each of these settings results in the highest responses for mean weight, thus confirming the parameters’ contribution. For the low mean weight observed in experiment 5, the settings of 2, 2 and 3 were used for L_h_, T and S (Table 2). The response for these print parameters confirms the lowest mean weight results, except for the T result (Table 5). However, as seen in the δ statistics, the T is the least important print parameter in terms of the contribution to optimum weight.

### 3.3. Tensile Testing Results

#### 3.3.1. Mean Results

In this study, an L16 OA was employed. For each combination of controlled parameters carried out, Tensile strength (σ), Stiffness (k) and Strain to failure (ε_f_), measurements were obtained. The results are presented in Table 6 and Figure 3.

The σ results show that the three factors influence the performance of the parts. In particular, experiment 3 has the lowest mean σ 13.28 MPa and experiment 4 has the highest at 20.46 MPa. This is a difference of >50.06%. The distributions are similar for experiments 1, 2, 3, 5 and 6 (StDev 1.15–0.87) and experiments 4, 8 and 9 have a more repeatable distribution (StDev 0.32 0.38). This shows that the chosen factors have an influence on this strength measurement.

The k results show that the three factors influence the performance of the parts in a similar fashion to the (σ) results. Experiment 9 has the lowest mean K 1.31 GPa, with experiment 3 at 1.32 GPa, like the result of experiment 4, has the highest mean result at 180 GPa. The difference between the highest and the lowest is >37.40%. The distributions are varied (StDev 0.033–0.123) and experiment 6 has the largest variability (StDev 0.123).

ε_f_ measurements differ from the k and σ results. Experiments 9 and 4 have the lowest mean ε_f_ with 0.12%, and experiments 1, 5 and 8 gave a result of 0.14%. The difference between the highest and the lowest is >16.6%.

The main difference from the k and σ results is the distributions. Experiments 1, 5 and 8 are highly repeatable, but the remaining experiments have a high variation. In particular, experiments 6 and 9 have the highest variability (StDev 0.013).

For all responses, the distributions are compared to the original PLA tensile results. To test the hypothesis (H_1_) that the LW-PLA parts have different tensile properties to the standard PLA parts, a paired *t*-test was performed (Table 7). The results for σ, k and ε_f_ show that H_1_ is correct. In all cases, the original PLA has higher results compared to the LW-PLA and there is a significant difference between the means.

#### 3.3.2. Print Parameters’ Contribution to Strength

An ANOVA was performed in order to assess the contribution of each processing parameter to the resulting strength results (Table 8). For σ and k, T is the most important process parameter, followed by L_h_ and S.

In Section 3.3.2, it was shown that experiment 3 has the lowest mean σ and k and experiment 4 has the highest σ and k. The lowest mean result settings used are 1, 3 and 3 for Lh, T and S, respectively, and the high mean result settings are 2, 1 and 2 (Table 2). The results, as seen in Table 8, confirm that each of these settings results in the highest responses for σ and k, thus confirming the parameters’ contribution.

The mean results shown in Figure 3 confirm that the three experiments with 260 °C temperature trend toward a below-average σ and k and the three experiments using the 240 °C setting are generally above the average. This is confirmed by the Difference (δ) result in Table 5, where, by reducing the T setting from 260 °C to 240 °C, the part σ strength increases by 24.2% and the k is increased by 16.5%. The S setting result showed to have very little influence on the σ and k. However, for ε_f_, it was the dominant factor. The results show that the print speed of 45 mm/s produces higher strain, while the 40 and 50 mm/s increase the ε_f_, with the change from 45 to 50 mm/s resulting in an increase of 6.6%.

Using an ANOVA general linear model (α 0.05) the significance of the responses for the selected factors is established. The results show that even though the selected factors and levels influence the mechanical properties of the test parts, the *p*-values (Table 8) show that the there is no statistically significant association between the response variables and the selected factors.

Using the DOE predicted results, it is possible to produce parts optimised for increased mechanical strength properties. Table 8 shows that the highest strength achievable for σ is 19.70 MPa. For K, it is 1.81 GPa and 0.145% for ε_f_. When the predicted responses are compared to the original mean responses for σ, K and ε_f_ (Table 7), an increased strength of 17.5%, 15.4% and 7.5%, respectively, can be achieved.

### 3.4. Optimisation for Part Weight and Strength

When designing lighter and more agile components, low-weight and high-strength components contribute to the overall performance and cost. The research has shown that the process settings have an influence on the part specifications and there are options to consider when selecting a particular criterion. In this section, contour plots are used to examine the relationship between the selected response variable of weight and three predictor variables of σ, k and ε_f_. With weight used as the fixed response variable, each contour plot can show two predictor variables (x- and y-axes).

With a focus on optimising a part for σ, ε_f_ and reduced weight, Figure 4a shows the following:It is possible to compromise with a high σ and ε_f_ but with an increased weight (8.35–8.40 g).For a low weight of 8.20 g, a medium σ of 16.90 MPa and high ε_f_ of 0.139% is achievable.For a low weight of 8.20 g, a high σ of 20.00 MPa and low ε_f_ of 0.120% is achievable.

When optimising a part for k, ε_f_ and reduced weight, the results are very similar to the previous observations (Figure 4b). The following can be achieved:A high k and ε_f_ can be realised but with an increased weight of 8.30–8.35 g.A low weight of 8.20 g is possible with a medium k of 1.62 GPa and high ε_f_ of 0.139%.A low weight of 8.20 g and a high k of 1.78 GPa is possible but with a compromise of a low ε_f_ of 0.123%.

For the optimisation of a part of σ, k and reduced weight, the results are different (Figure 4c).

The most important finding is that it is possible to achieve a low weight of 8.20 g together with a high σ of 20.16 MPa and k of 1.78 GPa.

## 4. Conclusions

Continuous efforts are being made to develop lightweight materials with improved stiffness, strength and energy absorption properties for a variety of multifunctional applications. Microcellular foamed polymers offer many advantages, including being lightweight, having improved mechanical properties, reducing material usage, providing better insulation and being cost-effective. These benefits make them an attractive material choice for a wide range of applications across various industries. This study reveals that microcellular polymers offer a solution for designers when there is a trade-off between part strength and weight. The key findings are as follows:

For all nine experiments, microcellular foaming was achieved and the LW-PLA parts are significantly lighter than standard PLA parts. The ANOVA was performed to assess the processing parameters’ contribution to weight, which found that L_h_ was the most important factor. Using the predicted result for this setting, it is possible to produce a part with a weight of 8.15 g. This is 21.22% below the average weight of a test part made from original PLA.

For each combination of controlled parameters, Tensile strength (σ) Stiffness (k) and Strain to failure (ε_f_) measurements were obtained. For all of the responses, the distributions are compared to the original PLA strength results. In all cases, the original PLA has higher results compared to the LW-PLA and there is a significant difference between the means. The chosen factors have an influence on each of these strength measurements. In particular, for σ there is a difference of >50.06% between the different experiments. For k, a difference of >37.40% is observed and ε_f_ measurements differ by >16.6%. This result shows the importance of optimisation during the manufacturing trials.

To improve manufacturability, the contribution of each processing parameter to the resulting strength results was investigated. For σ and k, T is the most important process parameter, followed by L_h_ and S. The mean results show that after reducing the T setting from 260 °C to 240 °C, the σ increased by 24.2% and the k increased by 16.5%. For ε_f_, the S setting is the dominant factor with a change of 6.6% measured.

Using the DOE predicted results, it is possible to produce parts optimised for increased mechanical strength properties. This approach identified that for σ, k and ε_f_, increased strengths of 17.5%, 15.4% and 7.5%, respectively, are achievable.

Optimisation for part weight and strength can be achieved. The compromise for reduced weight can be achieved with a medium σ and high ε_f_, and high σ and low ε_f_ is achievable. For k, ε_f_ and reduced weight, the results are very similar. For the optimisation of σ, k and reduced weight, it is possible to achieve a low weight with a high σ strength and a high k.

The use of microcellular polymers in FFF presents an opportunity to achieve parts with reduced weight and improved strength. Research in 3D printing is often centred on industrial applications. More recently, the uptake of this disruptive technology has moved beyond industry, and it is important not to underestimate the energy and material footprint that these machines have. The adoption of material-saving polymers could provide marked savings in energy consumption and lead to more sustainable manufacturing practices in 3D printing. The findings in this research are important for material selection and further research and development are needed to fully exploit the potential of microcellular polymers in 3D printing and enhance their application across different industries.

## Figures and Tables

**Figure 1 polymers-15-03910-f001:**
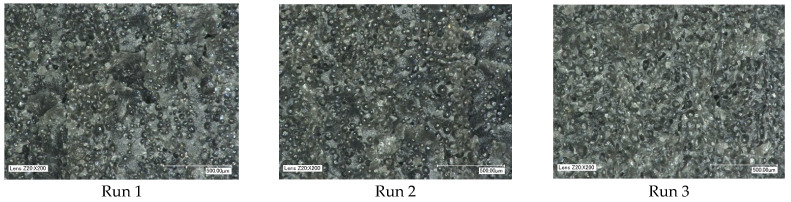
Cross-section of microcellular structures.

**Figure 2 polymers-15-03910-f002:**
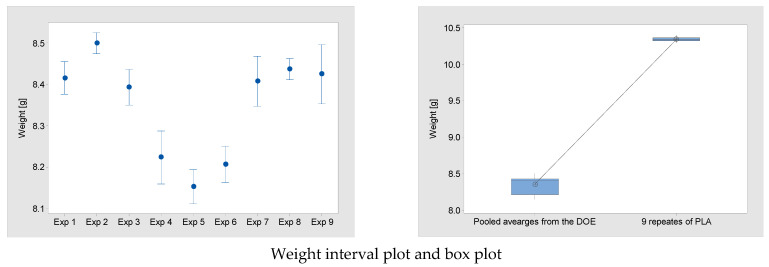
DOE results for part weight.

**Figure 3 polymers-15-03910-f003:**
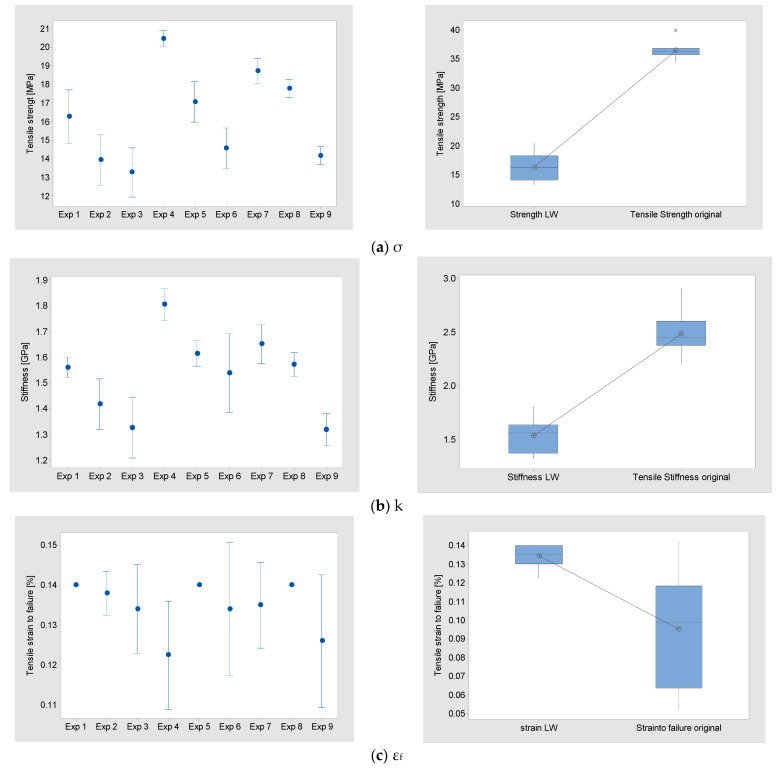
Interval plot and box plot.

**Figure 4 polymers-15-03910-f004:**
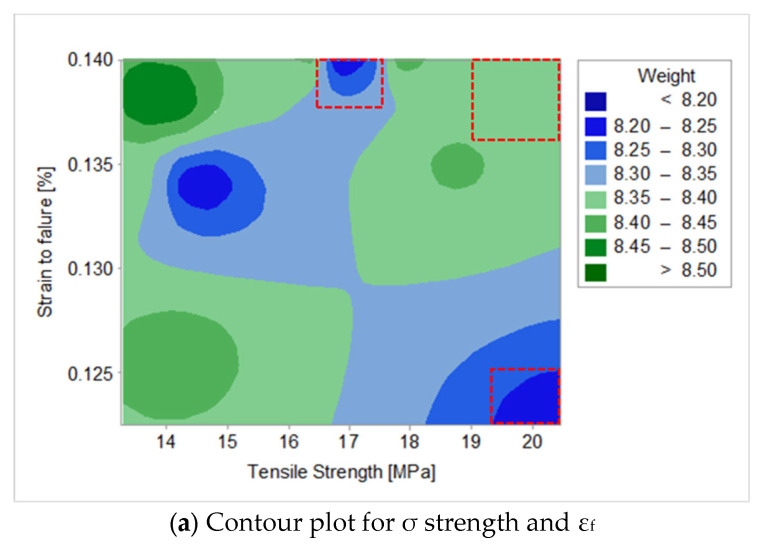
Control plots for optimisation.

**Table 1 polymers-15-03910-t001:** Default Ultimaker CURA Printing Setting.

	Default
Layer Height	0.2 mm
Nozzle Diameter	0.4 mm
Line Width	0.4 mm
Support Line Width	0.4 mm
Wall Line Count	2
Top Layers	4
Bottom Layers	4
Printing Temperature	230 °C
Build Plate Temperature	60 °C
Support Overhang Angle	45°
Connect Support Zig-Zags	On
Support Density	20%
Support Interface	Off
Build Plate Adhesion	Skirt
Skirt Line Count	3

**Table 2 polymers-15-03910-t002:** Taguchi L9 Orthogonal Array Design.

Run	DOE			L_h_ [mm]	T [°C]	S [mm/s]
1	1	1	1	0.28	240	40
2	1	2	2	0.28	250	45
3	1	3	3	0.28	260	50
4	2	1	2	0.24	240	45
5	2	2	3	0.24	250	50
6	2	3	1	0.24	260	40
7	3	1	3	0.18	240	50
8	3	2	1	0.18	250	40
9	3	3	2	0.18	260	45

**Table 3 polymers-15-03910-t003:** Part weight results.

Experiment	Mean W (g)	SE Mean	StDev
1	8.416	0.014	0.031
2	8.500	0.009	0.023
3	8.393	0.016	0.040
4	8.223	0.025	0.061
5	8.152	0.016	0.044
6	8.206	0.017	0.042
7	8.408	0.023	0.057
8	8.436	0.009	0.024
9	8.425	0.027	0.068

**Table 4 polymers-15-03910-t004:** Descriptive Statistics for Part Weight and Build Time H_1_: μ_1_ − µ_2_ ≠ 0.

Polymer	Mean	StDev	SE Mean
Part weight			
LW-PLA DOE Pooled	8.351	0.123	0.041
Original PLA	10.346	0.031	0.010
T-Value		*p*-Value	
−47.23		0.000	
Build time			
LW-PLA DOE Pooled	88.11	12.93	4.31

**Table 5 polymers-15-03910-t005:** Response for mean weight.

Level	L_h_	T	S
1	8.43	8.34	8.35
2	8.19	8.36	8.38
3	8.42	8.34	8.31
Difference (δ)	0.24	0.02	0.06
Rank importance	1	3	2
ANOVA *p*-Value	0.021	0.773	0.280

**Table 6 polymers-15-03910-t006:** Mean tensile results for LW-PLA.

Experiment	σ [MPa]	k [GPa]	ε_f_ [%]
1	16.26	1.56	0.14
2	13.94	1.41	0.13
3	13.28	1.32	0.13
4	20.46	1.80	0.12
5	17.06	1.61	0.14
6	14.56	1.53	0.13
7	18.74	1.65	0.13
8	17.78	1.57	0.14
9	14.18	1.31	0.12
Original PLA			

**Table 7 polymers-15-03910-t007:** Tensile Descriptive Statistics for H_1_: μ_1_ − µ_2_ ≠ 0.

Polymer	Mean	StDev	SE Mean
σ [MPa]			
LW-PLA DOE Pooled	16.25	2.45	0.366
Original PLA	36.41	1.41	0.45
T-Value		*p*-Value	
−21.64		0.000	
k [GPa]			
LW-PLA DOE Pooled	1.53	0.16	0.02
Original PLA	2.48	0.19	0.06
T-Value		*p*-Value	
−11.76		0.000	
ε_f_ [%]			
LW-PLA DOE Pooled	0.134	0.006	0.002
Original PLA	0.095	0.030	0.009
T-Value		*p*-Value	
3.98		0.003	

**Table 8 polymers-15-03910-t008:** Response for strength results.

Level	L_h_	T	S
σ [MPa]			
1	14.49	18.49	16.20
2	17.36	16.26	16.19
3	16.90	14.01	16.36
Difference (δ)	2.87	4.48	0.17
Rank importance	2	1	3
ANOVA *p*-Value	0.213	0.113	0.986
Predicted for high σ			19.70 [MPa]
k [GPa]			
1	1.435	1.671	1.556
2	1.652	1.534	1.513
3	1.513	1.394	1.530
Difference (δ)	0.217	0.277	0.043
Rank importance	2	1	3
ANOVA *p*-Value	0.090	0.059	0.723
Predicted for high k			1.81 [GPa]
ε_f_ [%]			
1	0.1373	0.1325	0.1380
2	0.1322	0.1393	0.1288
3	0.1337	0.1313	0.1363
Difference (δ)	0.0052	0.0080	0.0092
Rank importance	3	2	1
ANOVA *p*-Value	0.346	0.167	0.135
Predicted for high ε_f_			0.145%

## Data Availability

Data can be requested via the corresponding author.

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
