# Peer review of "Optimisation of 3D Printing for Microcellular Polymers"

_polymers, 2023, doi:10.3390/polym15193910_

Round 1
Reviewer 1 Report
This prensent study aims to demonstrate the appealing potential of 3D printing the foaming-based, light-weight PLA filament. While this is certainly an interesting topic, but the authors appear to have failed to present their results in an orgnized manner, which makes this paper rather confusing and difficult to understand. Therefore, the reviewer recommend the paper should be comprehensively modified before resubmission. The following is a few comments and suggestions.
1. The writing must be improved. There are more-than-accepatable errors through out the manuscript. For instance, "D printer" (line 151), "fig x" (line 237), "table x" (line 306), just to name a few. Therefore, the authors should carefully go-through the manuscript before resubmission.
2. The conclusion lacks innovation. It is rather a common sense that the original PLA will perform with higher mechanical strength than the foam-based, light-weight PLA (line 218) . What would be new, though, is to investigate how the printing strategies can manipulate the distribution and microstructures of the internal microcellular structure in the light weight PLA.
3. The experiment should be discussed more in-depth. It would be more straight forward if authors can provide detailed characterization on the presented light-weight PLA filament, such as density, microstructures, DSC, etc.
Overall, the quality of English language is accepetable.
Author Response
Rev1
This prensent study aims to demonstrate the appealing potential of 3D printing the foaming-based, light-weight PLA filament. While this is certainly an interesting topic, but the authors appear to have failed to present their results in an orgnized manner, which makes this paper rather confusing and difficult to understand. Therefore, the reviewer recommend the paper should be comprehensively modified before resubmission. The following is a few comments and suggestions.
- The writing must be improved. There are more-than-accepatable errors through out the manuscript. For instance, "D printer" (line 151), "fig x" (line 237), "table x" (line 306), just to name a few. Therefore, the authors should carefully go-through the manuscript before resubmission.
(Response) A thorough review has been done to remove the errors.
- The conclusion lacks innovation. It is rather a common sense that the original PLA will perform with higher mechanical strength than the foam-based, light-weight PLA (line 218) . What would be new, though, is to investigate how the printing strategies can manipulate the distribution and microstructures of the internal microcellular structure in the light weight PLA.
(Response) I would agree with the reviewer that the resulting microcellular structure would be interesting to analyse as a function of printing strategy. However, characterising density is impossible to quantify. Within this research x-ray analysis and SEM imaging of test piece cross section was undertaken. Unfortunately, both techniques proven inconclusive in characterising microcellular density. For x-ray analysis the length scale integration of microcellular cells distributed over a meso scale component made it very difficult to draw strong scientific conclusion. In addition, when analysing the cross section of the parts post fracture depending on the position of the break line different depths of cells would be exposed which again made post analysis impossible. I also agree that the resulting mechanical strength of foam-based PLA could be predicted. However, to the best of the authors knowledge no research has been conducted to the degree of mechanical strength reduction that would be witnessed as a result of the foaming characteristics.
- The experiment should be discussed more in-depth. It would be more straight forward if authors can provide detailed characterization on the presented light-weight PLA filament, such as density, microstructures, DSC, etc.
(Response) Please see point above for comments on density and microstructure. It would be interesting to characterise DSC, DMTA etc. However, this study focused on the tensile strength of foamed PLA as it was determined to be of more application relevance than other characteristics.
Reviewer 2 Report
Please provide all chemicals (with chemical structure) and material used in experimental part as materials section
1. Introduction is too long containing long paragraphs
2. Th aim is confused, what is the important of using of this kind of polymers and why the other thermoplastic polymers are not examined
3. What is the important of this polymer in the industrial field
4. What is the meaning of DOE and what is the different of DOE in table 2
5. What is the Taguchi L9
6. Have you used fuming agent to ensure foaming formation
Author Response
Rev 2
Please provide all chemicals (with chemical structure) and material used in experimental part as materials section
- Introduction is too long containing long paragraphs
(Response) The introduction has been edited resulting in a word reduction of 14%
- The aim is confused, what is the important of using of this kind of polymers and why the other thermoplastic polymers are not examined
(Response) As this paper focuses on FFF manufacturing, other thermoplastic polymers such as Polyethylene (PE), Polypropylene (PP), Polyvinyl Chloride (PVC), Polycarbonate (PC) and Polystyrene (PS) were not considered due to their inability to be extruded on consumer grade FFF printers. Others such as Polyethylene Terephthalate (PET), Acrylonitrile Butadiene Styrene (ABS) have been in the public domain for a number of years and have already been extensively researched and provided no additional value to this study. In the current pool of LW-PLA filaments available, there are only two manufacturers that manufacture PLA filaments that are considered foaming PLA, these manufacturers being eSun [1], and ColorFabb which is the focus of this study. Others market as “light weight” but no specifically foaming and as such as not valid in this study. Due to the material properties of these two brands of filament being very similar, the focus was put primarily on ColorFabb’s offering due to availability at the time of this study. However further comparisons could be made between these two filaments in future works [1] https://www.esun3d.com/epla-lw-product/
- What is the important of this polymer in the industrial field
(Response) This is an important point. We recognise that Lightweight polymers such as LW-PLA have several benefits that factor across various parts of the supply chain. Firstly manufacturing using these materials requires lower amounts of energy compared to their heavier counterparts, reducing costs and environmental impacts. The lighter weight of material means lowered powered equipment is required to produce and handle this material, potential simplifying the production process and again lowering costs. LW-PLA can produce the same functionality with less material than traditional PLA resulting in cost savings and reduced demand for raw. Additionally, the scope of use in increased due to regulatory and standards compliance in some industries that require products to meet certain weight criteria for safety and efficiency reasons. In industries such as aerospace and automotive, reducing the weight of components is crucial for improving fuel efficiency, performance, and overall sustainability. These points factor in to reducing transportation and storage costs, which overall reduces the environmental impact to the planet and aids in increasing sustainability across the manufacturing sector.
- What is the meaning of DOE and what is the different of DOE in table 2
(Response) DOE, or Design of Experiments, is a systematic approach used in research and manufacturing to efficiently explore and optimize various factors or variables that can influence a process, product, or outcome. By deliberately manipulating these variables and observing their effects, DOE helps uncover the relationships between factors and responses, enabling better understanding, prediction, and improvement of processes or systems. It's widely used to enhance efficiency, quality, and performance while reducing costs and resources in fields such as engineering. At our institute we find it a very effective way to uncover some of the more difficult relationships that we know to exist within a process.
5.What is the Taguchi L9
(Response) Following from above, there are different arrays that consider multiple factors and each having a different number of control levels. For our research we choose the L9 which is three control factors with three levels. We use this array for the performing of the experiments and the subsequent statistical analysis.
- Have you used fuming agent to ensure foaming formation
(Response) In this research the foaming component is within the polymer. Previous to our experiments we were unsure if our approach would result in foaming, but pre-trials and then the actual trials showed that foaming was achieved (figure 1)
Reviewer 3 Report
Building upon their recent research published in ref. [28], where they showed that printer and material modifications can significantly reduce polymer consumption by 55.8% to 56.4% and decrease power consumption by 29% to 38%, the Authors show, in the present manuscript, how to reduce part weight using a design of experiments approach when printing with a lightweight PLA filament incorporating a foaming agent. The Authors evaluate tensile strength (σ), stiffness (k) and strain to failure (εf) of the printed materials. The study allows the Authors to draw conclusions regarding the influence of the manufacturing process on weight savings and the observed relationship with material strength properties. The work is a bit routine-like, after the publication of [28], however it has its own merits, as the results well supporte the drawn conclusions and the utilization of the well-known method (not so often employed in the literature, though) of the design of experiments adds interest to the manuscript. The latter is written on a clear way and may be interesting for many readers. I recommend publishing it as it stands.
Author Response
Thank you for the positive response.
Reviewer 4 Report
This paper explorers the viability of manufacturing lightweight components using a Microcellular foaming polymer. And this text highlights the challenges and efforts in incorporating foaming techniques into 3D printing processes, specifically Fused Filament Fabrication (FFF). The description presented in this paper seems to be logical and sufficient. However, there are still some problems. I recommend the manuscript for publication in Polymers after revision. My comments are as follows.
1. In Lines 193-196, the authors mentioned that "The part production is highly dependent control factors, and therefore the effects of layer height (Lh), temperature (T), speed (S) and flow rate (F) have been investigated … was selected as shown in Table 2.". The authors first mentioned four factors Lh, T, S and F, but only three factors were mentioned in the following orthogonal test content, without F. Please explain why the F factor was not added to the orthogonal test.
2. In Line 207, the authors mentioned that "as presented in Table 3". The table serial number is incorrect. Please change Table 3 to Table 2.
3. In Lines 230 to 232, the authors mentioned that "The results show that the three factors influence the weight of the parts. In particular experiment 5 has the lowest mean W of 8.152 g and experiment 2 has the highest at 8.5 g." Please explain the influence of layer height (Lh), temperature (T) and speed (S) on the results of experiment 5 and experiment 2 respectively.
4. When the abbreviation of professional terms first appears, it needs to mark the full name, and the following full names are replaced by abbreviations. Due to the acronym "LW-PLA" appearing in the abstract, it is suggested to change "when printing with a Lightweight (LW) PLA filament incorporating a foaming agent." in Line 143 to "when printing with a lightweight PLA (LW-PLA) filament incorporating a foaming agent.".
5. In Line 265, the authors mentioned that "In particular experiment 3 has the lowest mean σ 13.28 MPa and experiment 4 has the highest at 20.46 MPa." Please analyze the reasons for the influence of three factors (layer height (Lh), temperature (T), speed (S)) on
experiment 3 and experiment 4. 6. Most of the references cited in this manuscript are relatively old. Please increase the number of references in the last two years.

Author Response
Rev4
This paper explorers the viability of manufacturing lightweight components using a Microcellular foaming polymer. And this text highlights the challenges and efforts in incorporating foaming techniques into 3D printing processes, specifically Fused Filament Fabrication (FFF). The description presented in this paper seems to be logical and sufficient. However, there are still some problems. I recommend the manuscript for publication in Polymers after revision. My comments are as follows.
- In Lines 193-196, the authors mentioned that "The part production is highly dependent control factors, and therefore the effects of layer height (Lh), temperature (T), speed (S) and flow rate (F) have been investigated … was selected as shown in Table 2.". The authors first mentioned four factors Lh, T, S and F, but only three factors were mentioned in the following orthogonal test content, without F. Please explain why the F factor was not added to the orthogonal test.
(Response) This is well spotted. The addition of F was an error, we have now removed the text.
- In Line 207, the authors mentioned that "as presented in Table 3". The table serial number is incorrect. Please change Table 3 to Table 2.
(Response) This has been rectified.
- In Lines 230 to 232, the authors mentioned that "The results show that the three factors influence the weight of the parts. In particular experiment 5 has the lowest mean W of 8.152 g and experiment 2 has the highest at 8.5 g." Please explain the influence of layer height (Lh), temperature (T) and speed (S) on the results of experiment 5 and experiment 2 respectively.
(Response) The section that you refer to is in 3.2.1. in the following section we cover the Print Parameters’ contribution to optimum weight the influence where layer height (Lh), temperature (T) and speed (S) is expanded. For continuity we decided that instead of adding further text to section 3.2.1, we added the following to section 3.2.2. ‘In section 3.2.2 it was shown that experiment 5 has the lowest mean weight and experiment 2 has the highest with a difference of 4.0% observed. The high weight observed in experiment 2 uses settings 1, 2 and 2 for Lh, T and S respectively (table 2). The results in Table 5 confirm that each of these settings results in the highest responses for mean weight, thus confirming the parameters contribution. For the low mean weight observed in experiment 5, the settings of 2, 2 and 3 were used for Lh, T and S (Table 2). The response for these print parameters confirms the lowest mean weight results, except for the T result (Table 5). However, as seen in the δ statistics the T is the least important print parameter in terms of the contribution to optimum weight.’
- When the abbreviation of professional terms first appears, it needs to mark the full name, and the following full names are replaced by abbreviations. Due to the acronym "LW-PLA" appearing in the abstract, it is suggested to change "when printing with a Lightweight (LW) PLA filament incorporating a foaming agent." in Line 143 to "when printing with a lightweight PLA (LW-PLA) filament incorporating a foaming agent.".
(Response) This has been done.
- In Line 265, the authors mentioned that "In particular experiment 3 has the lowest mean σ 13.28 MPa and experiment 4 has the highest at 20.46 MPa." Please analyze the reasons for the influence of three factors (layer height (Lh), temperature (T), speed (S)) on experiment 3 and experiment 4. 6.
Most of the references cited in this manuscript are relatively old. Please increase the number of references in the last two years.
(Response) The section that you refer to is in 3.3.1. in the following section we do cover the Print Parameters’ contribution to σ where layer height (Lh), temperature (T) and speed (S) is expanded. For continuity we decided that instead of adding further text to section 3.3.1, we added the following to section 3.3.2. ‘In section 3.3.2 it was shown that experiment 3 has the lowest mean σ and k and experiment 4 has the highest σ and k. The lowest mean result settings used are 1, 3 and 3 for Lh, T and S respectively and the high mean result settings are 2, 1 and 2 (Table 2). The results as seen in Table 8 confirm that each of these settings results in the highest responses for σ and k, thus confirming the parameters contribution.’
(Response) For the references it will come as a surprise that there are currently few manufacturing-based papers on the 3d printing of LW polymers. However, after a fresh review we have added seven recent papers to the submission.
Round 2
Reviewer 1 Report
Although authors have made significant efforts in updating the manuscript, there still are major flaws that need to be addressed before considering for publication.
1. Frankly, the discussion on the Figure 4 is still rather confusing. Figure 4b and c looks exactly the same (same legends, same plot, etc). although authors claim they should represent different things (one is stiffness to strain, the other is to stress). In addition, the way authors presenting the discussion and result is rather unorthodox (from Line 327 to Line 334), to say at least. As a reviewer, I cannot recall seeing a similar way to present the findings in a serious scientific journal. The authors should thoroughly polish this section.
2. There are a couple of places that require clarify. For instance, in Line 312, “For it is 1.81 GPa and εf 0.145%. Against the mean for each response the 312 increased strength is 17.5%, 15.4% and 7.5% respectively.” This sentence appears to be incorrect, and reviewer is not sure what the word “against” stands for. A more thorough proof-reading is strongly recommended.
3. In Figure 1, there appears to be very little difference between the sub-figures. I would recommend the authors to highlight the difference, and that is the purpose for comparisons.
Minor Revisions:
1. There are still several layout issues that need to be addressed. For instance, the “contour plot” in Line 321.
Requires a thorough proof-reading.
Author Response
Rev 1
Although authors have made significant efforts in updating the manuscript, there still are major flaws that need to be addressed before considering for publication.
- Frankly, the discussion on the Figure 4 is still rather confusing. Figure 4b and c looks exactly the same (same legends, same plot, etc). although authors claim they should represent different things (one is stiffness to strain, the other is to stress).
(Response). This is well spotted. The error was ours and has been resolved.
- In addition, the way authors presenting the discussion and result is rather unorthodox (from Line 327 to Line 334), to say at least. As a reviewer, I cannot recall seeing a similar way to present the findings in a serious scientific journal. The authors should thoroughly polish this section.
(Response)
This is a good point. The presentation of findings from contour plots are often quite problematic. Recognising this, we’ve simplified this section to three comments for figure 4a and 4b and one for figure 4c. We have also delineated the main observations with bullet points and added boxes in the figures to help the reader focus on the points observed. This makes the section much more readable.
- There are a couple of places that require clarify. For instance, in Line 312, “For it is 1.81 GPa and εf 0.145%. Against the mean for each response the 312 increased strength is 17.5%, 15.4% and 7.5% respectively.” This sentence appears to be incorrect, and reviewer is not sure what the word “against” stands for. A more thorough proof-reading is strongly recommended.
(Response) We agree this is not clear at all. The section is now re written for improved clarity.
- In Figure 1, there appears to be very little difference between the sub-figures. I would recommend the authors to highlight the difference, and that is the purpose for comparisons.
(Response) This is true. We use this figure to point out that the foaming process was achieved with all nine runs. We also say that due to the volume of bubbles it is not possible to discern the difference between the sum of bubbles and the population densities. We acknowledge that resulting microcellular structure would be interesting to analyse as a function of printing strategy. However, characterising density is impossible to quantify. So, this image just acts as proof that the microstructure was observed which we still believe is valuable to the goals of the paper.
Minor Revisions:
- There are still several layout issues that need to be addressed. For instance, the “contour plot” in Line 321.
(Response) This is now resolved, and a thorough proof-read edit has been completed.
Reviewer 2 Report
Dear Author
Thank you for your answering the reviewers comments
the article now can be accepted for publication
Author Response
Thank you for the review
Reviewer 4 Report
The authors revised the manuscript according to the previous comments and provided point-to-point replies. I think the manuscript can be accepted after minor revision. My comments are as follows:
Comments: The authors mentioned “Lightweight LW-PLA” and “Lightweight (LW-PLA)” in Lines 19 and 131. I recommend replacing them with “Lightweight PLA (LW-PLA)”.
Author Response
Thank you for the review